# Quinoa (*Chenopodium quinoa* Wild.) Seed Yield and Efficiency in Soils Deficient of Nitrogen in the Bolivian Altiplano: An Analytical Review

**DOI:** 10.3390/plants10112479

**Published:** 2021-11-16

**Authors:** Jesús E. Cárdenas-Castillo, José Delatorre-Herrera, Luisa Bascuñán-Godoy, Juan Pablo Rodriguez

**Affiliations:** 1Engineering Agriculture Department, Natural Sciences and Agriculture Faculty, Universidad Técnica de Oruro, Oruro 49, Bolivia; jecar65@hotmail.com; 2Doctoral Program in Agriculture for Arid-Desert Environments, Faculty of Renewable Natural Resources, Desert Agriculture Area, Universidad Arturo Prat, Iquique 1100000, Chile; 3Botany Departament, Faculty of Natural and Oceanographic Sciences, Universidad de Concepción, Concepción 4030000, Chile; lubascun@udec.cl; 4Julius Kühn Institute (JKI)—Federal Research Centre for Cultivated Plants, Institute for Plant Protection in Horticulture and Forests, Messeweg 11/12, 38104 Braunschweig, Germany; juan-pablo.rodriguez-calle@julius-kuehn.de

**Keywords:** quinoa, nitrogen harvested by yield, apparent use efficiency of N, arid environments, Altiplano

## Abstract

Quinoa is a strategic crop due to its high N content and its adaptability to adverse conditions, where most of the soils are deficient of nitrogen (N). The central question in this review was the following: How can quinoa yield low levels of nitrogen in the soils of Altiplano? This question was unraveled based on different factors: (1) fertilization effect on productivity, (2) fertilization limits, (3) uptake and assimilation of nitrogen parameters, (4) monoculture practice effect, and (5) possible sources and strategies. One hundred eleven articles of different scientific platforms were revised and data were collected. Information from articles was used to calculate the partial factor productivity for nitrogen (PFPN), the apparent use efficiency of N (APUE_N_), available nitrogen (AN), and nitrogen content harvested in grains (HarvN). Quinoa responds positively to fertilization, but differences in yield were found among irrigated and rainfed conditions. Quinoa can produce 1850 kg grains ha^−1^ with 50 kg N ha^−1^ under irrigated conditions, and 670 kg grains ha^−1^ with 15 kg N ha^−1^ in rainfed conditions. Quinoa increases seed yield and HarvN increases N fertilization, but decreases nitrogen efficiency. In Altiplano, without nitrogen fertilizer, the quinoa yield relies on between 500 and 1000 kg ha^−1^, which shows that in the soil, there are other nitrogen sources.

## 1. Introduction

Integrated nutrient management for food production is an approach and paradigm that supports the food security, conservation, and sustainability of renewable natural resources [1]. Understanding nutrient cycles is essential for improving crop nutritional management. Particularly, in highland and arid agroecosystems such as the southern Bolivian Altiplano, nitrogen (N) supply limits plant growth and development [2]. No other element for life, such as nitrogen, takes so many chemical forms in the atmosphere, soil, and plants [3]. In the atmosphere, the most reactive are N and gas (N_2_), while in soil, nitrogen oxide, NO, and nitrogen dioxide (NO_2_) prevail; when fertilizer is used, forms such as ammonia (NH_3_) can be found; while in water, nitrogen can be present in inorganic forms such as ammonia, ammonium, nitrate, and nitrite, and the organic form is present in proteins, amino acids, urea, and living or dead organisms [4].

In semi-arid and arid land regions, water resources are limited and have significant consequences on the soil nitrogen content [4]. The seasonal distribution of rainfall can affect the accumulation and emission of N in soils during the dry season [5,6]. Nitrogen is accumulated in the soil as wet and dry, and part of it is released to the atmosphere when pore spaces in the soil are filled with water, but this process depends on the soil type and climate [7].

Nitrogen use efficiency (NUE) determination in fragile soils such as the southern Bolivian Altiplano is significant for understanding soil NO_3_–N converted into grain for quinoa (*Chenopodium quinoa* Wild), a rainfed crop. NUE can be expressed in several ways: grain production by unit of available N, or index of utilization, which is the absolute quantity of produced biomass per unit of available N [8]. The factors that influence this efficiency are edaphic structure, climatic conditions, interactions between soil and bacterial processes, nature of organic and inorganic nitrogen sources, and availability of N in the soil [9,10]. NUE denotes the relationships between total input compared to the nitrogen output. This is complex and involves absorption, metabolism, and redistribution in the plant. However, adopting a complete crop nutrition strategy allows efficiency, profitability, and sustainability to improve. NUE is a determined metric used to measure N management in the soil [11]. Moreover, NUE is the maximum economic yield produced per unit of N applied, absorbed, or utilized by the plant to produce grain and straw [12]. NUE is partitioned in two processes: (a) absorption efficiency, when the plant is able to remove the available N from the soil usually present as nitrate or ammonium ions, and (b) utilization efficiency, when the plant is able to transfer the available N to the grain as protein [13]. The absorption efficiency is of the utmost importance for predicting plant performance and yield. Most plants capture inorganic N as dissolved nitrate (NO_3_^−^) or ammonium (NH_4_^+^) from the soil through their roots [14]. Root architecture, morphology, rate of respiration, and transporter activity for available forms of N in the rhizosphere determine N uptake rate. The utilization efficiency requires the process of carbon fixation for nitrogen taken up, photosynthesis, canopy formation, and nutrient remobilization from all tissues to grain during seed filling [10]. The process is initiated once N is introduced into the plant cell and is reduced into organic molecules.

Quinoa is a strategic partner crop for food security as a plant-based protein source [15], and its adaptability to unfavorable growing conditions [16,17]. Quinoa is an Amarantaceae with an intermediate protein content, less than that of legumes and more than that of cereals [17]. The protein content in grain depends on the varieties and soil conditions, and it can reach up to 23%. This protein level requires a significant supply of nitrogen, which is not only essential for the grain, but also for plant growth and development. Its exceptional adaptations to limiting factors in the environment are tolerance to drought [18] and frost [19], as well as to saline and/or low-fertility soils, maintaining adequate yields [20,21]. The Intersalar region, in the southern part of the Bolivian Altiplano has an extreme climate, with rainfall from 150 to 300 mm per year, 200 days of night frost, strong winds, and intense solar radiation [22,23,24]. Furthermore, it has serious degradation problems and low nitrogen levels in the soil. Soil degradation is attributed to monoculture, the use of virgin soils to expand the agricultural frontier, the use of inadequate agricultural machinery (disc plough) in highly susceptible soils to wind erosion, traditional and manual harvesting, the use of left-harvested plants after grain threshing in camelid cattle, llama (*Lama glama*) or sheep (*Ovis aries*) feeding, neglect of traditional sowing in the traditional system of sectoral fallowing known as “mantos” [25], the practice of soil fallowing (two to three years without agriculture), and the lack of organic matter due to little or no incorporation of manure (reduction of llama and sheep livestock) and stubble leftover.

In the Intersalar, it has been observed that there are plots with more than 80 agricultural years of production under quinoa monoculture. There are no contributions of manure or other nitrogen mineral fertilizer sources, and the only form of cultural management is the practice of soil fallow (one to two years) [26]. However, acceptable and economically sustainable productivity is still utilized by farmers, and the yields are between 450 and 750 kg ha^−1^, despite no application of nitrogen [27,28,29,30]. Due to their origin, and as a consequence of the abovementioned factors, the soils of the Intersalar zone are poor in N. Of these soils, 98% are classified as very low in N, while the remaining soils (2%) are classified as low [26,31]. With this position, we ask ourselves, how can quinoa be produced in the Bolivian Altiplano under low levels of nitrogen in the soil? This question was unraveled based on different factors: (1) the effect of fertilization on productivity under rainfed and irrigated agricultural conditions, (2) the top and bottom limits of fertilization, (3) the parameters related to the uptake and assimilation of N, and (4) the effect of monoculture on yield under rainfed agricultural conditions. This article aimed to explore theoretically the efficiency of nitrogen under rainfed and irrigated cultivation conditions to allow us to explain the performance of quinoa in Bolivian Altiplano soils without the application of fertilizer.

## 2. Results

### 2.1. The Effect of Fertilization on Productivity in Rainfed and Irrigated Quinoa Cultivation

Nitrogen fertilization in quinoa is an unsolved issue; the literature data show great variability in results, ranging from very low application (30 kg of N and production of 5.5 tons of grains per ha^−1^ [32]) to high application (175 kg ha^−1^ with 4.2 tons of grains per ha^−1^ [33]) (Table 1). The variation in the results regarding nitrogen fertilization over seed yield expressed in Table 1 are the consequences on cropping conditions, varieties, soil textures, irrigation systems, and management.

Table 1 shows the performance of several quinoa varieties cultivated even under the same nitrogen fertilizer rate, which influenced seed yield under rainfed and irrigated conditions. We can see in the table that not only did the nitrogen fertilization improve the seed yield, but irrigation, water quality, organic matter content, pests, weeds, planting density, and varieties also strongly influenced the seed yield. The average data for each dose of fertilizers presented in Table 1 were utilized to calculate and depict seed yield variation under irrigated and rainfed conditions in Figure 1.

From the data obtained, we can infer that these conform to a normal curve, with an increase in yield up to 240 kg ha^−1^ for applications of nitrogen fertilizer, after which it begins to become asymptotic and the yields begin to decrease after 300 kg ha^−1^ nitrogen fertilizers. The relationship between nitrogen uptake and nitrogen fertilizer rate (*R*^2^ = 0.64, *p* = 6.7 × 10^−8^) was remarkably consistent (Figure 1).

### 2.2. What Happens in the Altiplano Agroecosystem with a Quinoa Monoculture and without Nitrogen Fertilizer Applications under Rainfed Conditions?

The previous data and figure presented in the above section support the possibility that the performance of quinoa can increase under rainfed conditions, e.g., in the southern Bolivian Altiplano, which is an agroecosystem featuring sandy soils and a very low nitrogen content [26], with minimal use of manure and no chemical fertilizers. Figure 2 depicts the intensity of land use through several growing seasons in a plot exclusively for quinoa cultivation. The N available or contained in the soil decreased for the 2017 growing season. The curve shows the relationship of a monoculture practice in the same plot for several years, which decreased the nitrogen in the soil (*R*^2^ = 0.63, *p* = 1.2 × 10^−4^) in the Intersalar area of the southern Bolivian Altiplano [26]. These data explain that in 90.9% of the farms, and according to the FAO (2013) [43], the soil nitrogen content is classified from very poor to poor (less than 0.15% N), with only 9.1% of the soils having a medium content (0.15–0.25% N).

Figure 2 shows the dynamics of total soil nitrogen and seed yield for several growing seasons. The data and curves presented in Figure 2 are based on 113 plots, with different years of quinoa monoculture use (*R*^2^ = 0.41, *p* = 0.009), and only soil fallow of the plot without incorporation of organic matter or fertilizers and rainfed irrigation have been practiced [26]. Quinoa cultivation in the plots is practiced through the traditional Aymara system known as “mateado” or “speckled” and planted in holes (“kollas”), with planting densities of 1 × 1 m and three to four plants per “kolla” (i.e., planting rate). Figure 2 shows higher yields than 500 kg ha^−1^; however, 60.9% of the farms evaluated do not exceed 1000 kg ha^−1^, which is surprising due to the low nitrogen content of the soil (Figure 2), not exceeding 0.10–79.6% in the farms. Quinoa cultivated in the southern Altiplano of Bolivia, as through the case in Figure 2, denotes that in monoculture practice, the total soil nitrogen decreases over time, while seed production does not disappear, even though soil has a very low nitrogen content.

When associating N values to the yield (Figure 3), it was found that in the trials with nitrogen fertilizer application, the curve trend showed an increase in yield at a higher concentration of nitrogen in the soil, which is asymptotic starting from 0.14% of total soil nitrogen.

Figure 3 shows similar trends to Figure 2 for quinoa with fertilization, although Figure 3 depicts higher total soil nitrogen (*R*^2^ = 0.47, *p* = 0.00792); however, the seed yield increased until it stabilized, and then it decreased. When analyzing the nitrogen use efficiency indicators (Table 2), deterioration can be observed.

According to Table 2, quinoa is more efficient at using nitrogen when there is less N in the soil. For example, in soils with very low nitrogen values such as 14.9 kg ha^−1^, 45.1 kg of grain can be obtained for each kilogram of nitrogen, meaning an APUE_N_ of 122.8%, which is difficult to explain without considering an extra contribution of nitrogen to the crop. However, soils with 0.22% of total soil nitrogen or 163 kg of available nitrogen produce only 4.8 kg of grain for every kilogram of available nitrogen.

## 3. Discussion

### 3.1. The Effect of Fertilization on Productivity in Irrigated and Rainfed Cultivation

The average data for each dose of fertilizer presented in Table 1 were used to determine the nitrogen uptake, expressed as kilograms of N to produce one ton of grain, as described in Figure 4. The relationship between nitrogen use efficiency and nitrogen fertilizer rate was remarkably consistent (*R*^2^ = 0.88, *p* = 2.2 × 10^−4^).

From the data obtained, we can infer a normal curve, with an increase in nitrogen uptake per ton of produced grain, up to applications of 260 kg ha^−1^, after which it started to become asymptotic, and over 400 kg ha^−1^, the yields began to decrease as further nitrogen fertilizers were incorporated. In Figure 4, we demonstrate that nitrogen uptake increased when the nitrogen fertilizer rate increased from 35 kg of N per ton of with no nitrogen fertilizer, until reaching the optimum 90 kg of N per ton of produced grain with 260 kg N ha^−1^. Alvar-Beltran et al. used three doses of nitrogen fertilization (25, 50, and 100 kg N ha^−1^) and the extraction was 25 kg N per ton of grain produced (1:40 ratio) [44].

### 3.2. The Limits of Fertilization in Quinoa

Table 3 shows the efficiency indicators based on the average yields from each nitrogen fertilization rate, according to the data and average values in Table 1.

By increasing the nitrogen fertilizer rates, the seed yield increased, reaching an optimum production at 130 kg of N ha^−1^. However, after this point, the seed yield decreased, as depicted in Figure 5. Similar results were found in trials with a quinoa genotype O3 and two other cultivars [39,45]. The results were adjusted, with a good correlation to the law of diminishing returns (*R*^2^ = 0.83, *p* = 0.00513) [45] and agreement with Pandey et al. [46], who indicated that high rates of nitrogen in crops cause a depressive effect.

Figure 5 depicts the break-even point at which quinoa is efficient enough under certain levels of nitrogen fertilization. Based on the data in Table 3, Figure 5 shows a higher efficiency in the use of nitrogen in quinoa grown in soils even under very low nitrogen levels. A balance point appeared, indicating that up to 130 kg ha^−1^ of nitrogen is enough to produce 2700 kg ha^−1^. This point is a balance between APUE_N_ and seed yield, even when quinoa is grown with high rates of nitrogen fertilizer. There is a remarkable relationship between decreasing APUE_N_ and an increasing amount of nitrogen fertilizer (*R*^2^ = 0.77, *p* = 0.0012).

### 3.3. Parameters Related to the Uptake and Assimilation of Nitrogen

The PFP_N_ will depend on the physiological efficiency of the cultivar, that is, the proportion of available N absorbed by the crop, and on the losses during the cycle [43]. Nitrogen use efficiency averages 33% in cereals, indicating significant potential for improvement [47]. For quinoa grown on dry land and without an extra contribution of nitrogen fertilization, the PFP_N_ decreases when the nitrogen content in the soil increases (Table 3). With values for PFP_N_ ranging from 59.6 for very nitrogen-poor soil (0.02%) to 6.3 for very nitrogen-rich soil (0.22%), this means a loss of 89.4% in N efficiency.

In a trial with quinoa and five levels of N (0, 40, 80, 120, and 160 kg N ha^−1^), the highest PFP_N_ was recorded with 40 kg N ha^−1^ and 30.52 kg of grains produced per kg of N applied [13]. Another study showed that the efficiency of the use of nitrogen in the yield of quinoa with N doses of 0, 50, 100, 150, and 200 kg N ha^−1^ was affected by higher availability of N in the soil [42]. The data in Figure 2 show a deterioration in PFP_N_ and APUE_N_ when higher doses of fertilizers are included, although the yields increased. For example, applications of 40 kg N ha^−1^ produced 52 kg of grain for each kilogram of fertilizer applied. This is in contrast to doses of 160 kg N ha^−1^, where only 17 kg of grain were produced for each kilogram of fertilizer applied, and with 400 kg N ha^−1^, which produced only 5 kg of grains for each kilogram of nitrogen, that is, a 90.4% efficiency deterioration. The apparent use efficiency of nitrogen (APUE_N_) shows that the applied or available nitrogen was multiplied 1.4 times in harvested grains. Differently, when 400 kg N ha^−1^ was applied, only 0.145 times the applied dose was harvested. The 1.4-fold increase in nitrogen content is striking, which could be explained by the presence of microorganisms or the contribution of rain, or by the deepening of the roots to increase the volume of soil to explore.

In addition, Figure 5 shows the balance point between yield and APUE_N_, which means that the optimal ratio of yield versus nitrogen dose was around 130 kg N ha^−1^.

The data obtained are consistent with those of Franco Alvarado (2018) [41], who applied up to 200 kg N ha^−1^, finding that the absorption efficiency use of nitrogen (APUE_N_) decreased as the applied dose of N increased. Without the application of nitrogen fertilizer, it reached the highest APUE_N_. In contrast, upon application of 200 kg N ha^−1^, the seed yield decreases. Franco Alvarado [41] found that the optimal dose of available N (62 kg N ha^−1^) in the soil achieved the highest productivity in quinoa crop. This deterioration in the efficiency indicators indicates that increasing application of nitrogen fertilizers in quinoa is not used to produce grains, it could be derived from the production of biomass [48], or else there is a significant loss of this element by leaching. It has been estimated that between 50% and 70% of the applied nitrogen is lost from the soil–plant system, by surface runoff or leachate or by microbial denitrification, a process by which nitrate is converted to nitrogen oxides (N_2_O and NO) and elemental nitrogen (N_2_) is also lost by volatilization [44]. The loss of N by drainage (19.7 g N m^−2^) represents the main output and the volatilization of urea (8.65 g N m^−2^) [17].

The efficiency in nitrogen uptake and transfer to grains (APUE_N_) explains the total nitrogen harvested in the grain compared to the total nitrogen uptake per ton of grain. Table 3 shows that plants with nitrogen deficiency stress have a higher APUE_N_. The quinoa plants used the little available nitrogen better to produce grains, with a lower yield. The nitrogen-deficient plants showed a decrease in aerial and root biomass and a lower seed yield, but a greater efficiency in the use of nitrogen. Similarly, Calvache and Valle [48] found that as nitrogen increases, the aboveground biomass also increases (Table 4).

Our data resemble those of Alvar-Bertran et al. [44], who compared height and canopy in plants with seed yield. The highest seed yield was concentrated in plants of 40–60 cm with a 3–5% canopy. Calvache and Valle [48] compared the biomass produced by quinoa and seed yield as a function of the nitrogen dose under irrigated or rainfed condition (Figure 6). Unfortunately, the data only reached doses of 150 kg ha^−1^, which did not allow one to establish, in higher doses, what the real behavior would be. Figure 6 shows that as the dose of nitrogen increased, the production of biomass also increased, while under rainfed and irrigation conditions, the rate of biomass production decreased.

Higher doses of nitrogen were derived by the quinoa plants to increase the above vegetative growth rather than to grain production (Figure 6), while decreasing the efficiency of nitrogen for grain production.

The accumulation and redistribution of N are important for the yield and quality of grain [49]. The supply of quinoa grain, like all seed-producing plants, depends on the N accumulated before anthesis. In wheat, approximately 50–95% of the N in grain at harvest comes from the remobilization of the N stored in the shoots and roots before anthesis [49,50]. In quinoa, these values have not been determined; however, data from Calvache and Valle [48] are depicted in Figure 7, where the nitrogen content in the panicle begins to increase from 80 days after sowing, while it remains the same in the stems and decreases in the leaves.

In all vegetable species, plant stress accelerates senescence and alters the source–sink relationship, resulting in a significant reduction in crop yield [51]. The southern Bolivian Altiplano has in average of 237 mm of annual rainfall (SENAMHI, Bolivia [52], (www.senamhi.gob.bo, accessed on 15 January 2021), and this is not enough for the cultivation of quinoa. Plants grow under conditions of water deficit, which, added to other environmental factors, creates a condition that accelerates plant senescence because of stress. Naturally, senescence induces the cessation of vegetative growth, accelerates the flowering and fruiting process, changes the plant metabolism, and alters the redistribution and partition of nutrients [53]. Stress senescence affects agronomic characteristics, including the efficiency and yield of carbohydrate/nitrogen use (C/N) and the C/N balance in the source–sink relationship [54,55]. The nitrogen remobilization efficiency (NRE; corresponds to the proportion of N in the crop) depends on the amount of N remobilized to the grain in the period after anthesis and the amount of N stored in the vegetative parts during anthesis. It is important to guarantee that stress senescence has not started prematurely, as nitrogen transportation into the grain will be affected [56]. After the plant takes up nitrogen and metabolizes it into plant proteins, this nitrogen is remobilized to the developing grain [57,58,59].

The growth of fruits and seeds indicates a new sink that competes with the rest of the plant for nutrients. At this point, the nitrogen partition process is important. Bascuñán-Godoy et al. [60] found that the total protein content in quinoa decreases with stress and increases when irrigated again. This decrease correlates with increases in NO_3_^−^ and NH_4_^+^. The increase in NO_3_^−^ could be associated with a marked stress-induced decrease in nitrate reductase (NR) activity, and the increase in NH_4_^+^ is probably associated more with the improvement in the protein degradation and re-assimilation processes of N [61]. Although, it can also be associated with the availability of water, which allows mobilization in the soil to the rhizosphere, improving nitrogen absorption and the presence of microorganisms that provide nitrogen to the plant.

### 3.4. The Effect of Monoculture on Yielding in Non-Fertilizer Rainfed Cultivation in Bolivian Altiplano

Our study demonstrated how a low soil nitrogen content, as in the southern Bolivian Altiplano, is associated with similar studies [44]. Of the Intersalar soils in the southern Bolivian Altiplano, 91% are sandy loam and sand [26]. The soil texture affects the availability of N by inducing the mineralization and the depth and distribution of the rooting system. Therefore, the application of 120 kg N ha^−1^ in plots with different soil textures results in differences in nitrogen absorption in quinoa: 134 kg ha^−1^ for sandy clay loam, 102 kg ha^−1^ for sandy loam, and 77 kg ha^−1^ for sand under full irrigation [20]. This situation is important, since the N from deep soil can be absorbed by diffusion and is an important part of the total absorption [62]. Based on applications of 25, 50, and 100 kg N ha^−1^ in a quinoa crop in Burkina Faso, Alvar-Beltran et al. [44] determined that the nitrogen concentration decreases from 0.051% to 0.037%, in depths from 20 to 60 cm, respectively, for applications of 25 kg ha^−1^, while for 100 kg ha^−1^, it decreases from 0.035% to 0.029%.

Under adequate water conditions, the quinoa seed yield increased with higher doses of N, as well as the harvest of N per hectare. In Figure 3, we show that the amount of N absorbed in the grain increased to 90 kg of N per ton of grain produced with 240 kg of nitrogen fertilizer, but the APUE_N_ decreased from 100.5% with 50 kg of nitrogen fertilizer to 37% with 240 kg of nitrogen fertilizer (Table 3). When analyzing how the nitrogen deficit affects the conditions of the southern Bolivian highlands, from Table 2, it can be seen that very low values of soil total N (0.02%) equivalent to 14.4 kg N available ha^−1^ produce average yields of 670 kg of grain ha^−1^ with an APUE_N_ of 122%, which is surprising, since it contained 22.8% more nitrogen than that provided by the soil. The maximum yield point was obtained with 0.13% of total N in soil, equivalent to 96.5 kg of nitrogen ha^−1^; with this amount, 1866 kg grain ha^−1^ was produced, but the APUE_N_ decreased to 52.6. However, these values are close to the equilibrium point shown in Figure 5 for fertilized and irrigated crops (130 kg N ha^−1^). These values agree with those of Cassman et al. [57], who showed that a low content of N in the soil contributes to an increase in the efficiency of N.

### 3.5. Sources and Strategies to Improve N Supply and Efficiency in Quinoa in Non-Fertilized Soil in the Altiplano

#### 3.5.1. Sources

Another source of soil N content comes from the atmosphere, which contains 79% by volume of nitrogen, making it a source of great reserve for the system, since it feeds the nitrogen cycle. In the Bolivian Altiplano, there is no history or records about the contribution of nitrogen by rainwater and the atmosphere [63]. The rainfall in the 2016 and 2017 seasons was between 194 and 280 mm year^−1^ (SENAMHI, Bolivia, www.senamhi.gob.bo, accessed on 15 January 2021). This low level of rainfall makes it difficult to evaluate the contribution of N. There are many controversies about the amount of N deposited through this way in soil. In temperate climates, it can fluctuate between 0.74 and 21 kg N ha^−1^ year^−1^ [64], and 15 kg N ha^−1^ year^−1^ [65] could be considered the average. These amounts would be higher in tropical climates, i.e., between 6.5 and 72 kg N ha^−1^ year^−1^ [64].

In the Bolivian Altiplano, electric shocks can be intense [66], but the amount of rain is much less. It is unknown whether the factor of electrical discharge and/or static electricity results in a higher nitrogen contribution, or why a scarcity of nitrogen is observed in Altiplano soils, which is significant in the soil nitrogen balance [63]. The electrical discharge that occurs during storms synthesizes nitrogen oxides from nitrogen (N_2_) and oxygen (O_2_) in the air, being driven into the ground by rain [67,68]. The quantity of nitrate produced across the world is estimated to 7.5 million ton per year.

#### 3.5.2. Strategies

Interestingly, other parameters such as root biomass only correlate with the seed yield under low nitrate conditions, but not at sufficient levels of nitrate [69]. It has been published that root biomass is not important for the uptake of N [70], or even that plants cannot uptake N during grain filling [71]. However, Mi et al. [72] reported that root biomass is an important attribute for N uptake in corn at low nitrate levels (but not at sufficient nitrate levels). The roots of maize can take up N even during the reproductive phase [10]. Coke and Gallais [73] estimated that 62% of the N in the kernel originates from N remobilization, and 38% is derived from post-silking root N uptake. Recently, it has been reported that the increase in the number of secondary roots is related to the upregulation of nitrate transporter gene (Cq*NRT2*) under low nitrate conditions in quinoa seedlings of both Socaire (an Andean landrace) and Faro [74]. This indicates that a low amount of N induces in quinoa a series of mechanisms to cope with low N.

There are antecedents that relate nitrogen deficiency with other active compounds such as Strigolactones (SL). These hormones act by activating the signaling pathways that allow lipid catabolism to be the main carbon source in fungi. Under nutrient deprivation conditions, the production of large amounts of SL leads to the suppression of shoot branching and stimulates symbiosis [75,76]. Strigolactones promote the modification of the architecture of roots and shoots and stimulate a symbiosis of rhizobia bacteria and AMF fungi, and SLs play a crucial role in nitrogen and phosphorus deficiency.

Another of the strategies used by halophytes to capture nutrients is the association with soil microorganisms, especially arbuscular mycorrhizal fungi (AMF), which promotes growth and development under stressful conditions [77,78,79], and plant growth-promoting rhizobacteria (PGPR), with the ability to colonize the roots of many plant species, contributing to their development and survival [44].

The participation of arbuscular mycorrhizal fungi (AMF) in quinoa, a facultative halophyte, is debatable, since the presence of root symbiont fungi in Bolivian Andean quinoa plants is insignificant [80], and plant growth responses could be considered a mutualism–parasitism continuum [81]. However, some research, e.g., in the desert zone of Chile, has determined that there is a high presence of mycotrophic plant species with a high variation in the degree of mycorrhization in the root (mycorrhizal colonization and the mycorrhizal medium), through the production of resistance spores and extraradical mycelium [82]. Despite the low level of AMF colonization, it has been proposed that quinoa could be an interesting component for crops rotation to improve and increase N cycling in soils compared to other crops [83].

In quinoa, in particular, there are very few investigations on the presence of fungi and their contribution to growth or to withstand stressful conditions. The dominant fungal genera (*Penicillium*, *Phoma*, and *Fusarium*) have been detected in the roots of quinoa [84]; for example, Macia-Vicente et al. [85] and Khan et al. [86] previously found them as root inhabitants in several plant species. These fungal genera play a positive role in plant growth and tolerance to abiotic stress. The endophyte fungus community has been recognized as one of the Chilean quinoa ecotypes [84]. Despite a relatively high diversity of endophytic root fungi associated with quinoa plants, the dominant fungal community consists of only *Ascomycotaphyla*. The most abundant fungal genera in quinoa are *Penicillium*, *Phoma*, and *Fusarium*, which are common endophytes in plant roots, highlighting endophytic root fungi as a new additional performer [85].

Furthermore, there is a history of the participation of bacterial endophytes associated with quinoa [85,86]; 100% of quinoa seeds are inhabited by several bacteria from the genus *Bacillus* [85], which probably induces a state of natural readiness in quinoa plants, allowing them to overcome extreme environmental situations. Among the best-known microorganisms with PGPR activity are species of the genera *Rhizobium* sp., *Azospirillum* sp., and *Pseudomonas* sp. [87,88]. There are several mechanisms by which bacteria contribute to the germination, growth, and survival of plants, including biological nitrogen fixation, solubilization of phosphates, production of siderophores, biosynthesis of phytohormones (auxins, cytokines, and gibberellins), synthesis of antibiotics, and induction of systemic resistance [89,90]. Under low nitrogen concentrations, auxin biosynthesis and transcriptional accumulation are induced, thus regulating lateral root formation. Conversely, lateral root growth can be inhibited with a higher than optimal supply of N [91,92]. Nitrate transporter (NRT) genes have also been reported to be responsible for the high affinity of the NO_3_^−^ transport system, which is related to the growth of lateral roots [74,93,94]. Based on the information provided, it is possible to assume that part of the nitrogen supply of quinoa in conditions of deficit of this element, is supplied by the interaction that occurs with these microorganisms.

## 4. Material and Methods

A systematic review (SR) was used to identify, select, and critically appraise the relevant primary information. The data included several studies and information extracted and analyzed according to the proposed methodology [95]. Throughout this review, the primary question was “How does quinoa obtain the necessary nitrogen for production under rainfed conditions with nitrogen deficiency?”

The databases Google Scholar, EBSCO, ISI Web of Science, Scopus, Elsevier Science Direct, Oxford Journals, Wiley Online Library, Springer, and Nature were used. Furthermore, project reports, university theses from national and international organizations such as FAO, and World Bank were used to find information. The keywords used were “quinoa,” “yield,” “Intersalar,” “southern Altiplano,” “efficient use of N,” and “fertilization with N.” The analysis focused on the Intersalar zone in the southern Bolivian Altiplano. A total of 1060 abstract articles were downloaded and read, and then 125 were reviewed, before finally selecting 54 articles. Finally, 10 articles were considered to determine the fertilization rates, as presented in Table 1. The average, standard deviation, and standard error values were calculated with INFOSTAT v2020 [96], and regression linear and scatter plots with OriginPro v2019 (OriginLab Corporation, Northampton, MA, USA) [97].

The collected data were evaluated with the indicators nitrogen use efficiency (NUE), partial factor productivity for nitrogen (PFP_N_, kg grain kg^−1^ N_fert_), and apparent use efficiency of N (APUE_N_,%) according to the methodology described by Zhang et al. [98].

PFP_N_ is the ratio of crop yield per unit of nitrogen fertilizer applied by Zhang et al. and Kuosmanen [99,100], or that available in the soil, as per Equation (1):PFP_N_ = Y_g_/N_fert_,(1)
where Y_g_ and N_fert_ are the grain yield (kg ha^−1^) and N_fert_ is the application rate of nitrogen or the available nitrogen (NA) under non-application of fertilizer conditions.

APUE_N_ was calculated according to the recommendation by Al Naggar et al. [101], as per Equation (2):APUE_N_ = (N_kg_/N_fert or NA_) × 100(2)
where N_kg_ is the amount of N in quinoa grains (kg N grains ha^−1^).

In soils without fertilizer, the data of total nitrogen (N_t_, %) were used to calculate the available nitrogen (AN), and the results were transformed to kilograms per hectare (kg ha^−1^) [102]. A weight of 1 ha of soil was determined, with a depth of 30 cm and an apparent density of 1.65 t m^−3^. The density of the soils in the Intersalar zone is related to its texture. Therefore, 91% of the soils had sandy loam and sand texture with a bulk density of 1.4–1.9 g cm^−3^, with a mean of 1.65 g cm^−3^. For the estimation of AN, Equation (3) was used:AN = Nt × 0.015,(3)

Soil nitrogen has two components, organic N and inorganic N. Soil microorganisms convert organic forms to inorganic forms, which plants absorb. It has been estimated that between 1.5% and 3% of the total N in the soil corresponds to inorganic N. We worked with 1.5% (or 0.015); with this, the available nitrogen (AN) was determined following the methodology of Aguilar et al. and Castellanos et al. [103,104].

The extracted or harvested nitrogen content (Harv_N_) was estimated using an average value in quinoa of 16.6% protein, according to data taken from Abugoch et al. [105] and Covarrubias et al. [106], with a protein content of 16% nitrogen [107] and a conversion factor of 0.0272 used for the calculation of Harv_N_. The nitrogen content harvested in grains (Harv_N_) was determined with Equation (4):Harv_N_ = Yield grain × 0.0272(4)

## 5. Conclusions

Globally, quinoa cultivation is of interest due to its versatility, resilience, and provider of a nutritious grain. Nevertheless, quinoa cultivated in Andean countries, particularly in Bolivia are facing several constraints as low fertility in soils. Data obtained from this review provided better knowledge about increasing yield of quinoa (i) the optimal amount under these conditions was determined to be 240 kg of N with 3600 kg grain ha^−1^ (ii) the fertilization limits were obtained from the intersection of APUE_N_ and seed yield, finding that the equilibrium point appeared to be up to 130 kg ha^−1^ of nitrogen, which is enough to produce 2700 kg ha^−1^, (iii) the nitrogen uptake and assimilation parameters presented an inverse relationship, with higher doses of fertilizers showing a lower efficiency of nitrogen utilization in grain production. The information obtained shows that in the event of an increase in nitrogen fertilization, an important part of the N is destined for vegetative growth. (iv) Under monoculture practice without nitrogen fertilizers, as occurs in the Bolivian Altiplano, the response found in terms of yields is similar to the practice of irrigation and fertilization, although in much smaller proportions, in soils with very low nitrogen (0.02% total N), equivalent to 14.9 kg ha^−1^ N available. Meanwhile, 670 kg grains ha^−1^ and 45.1 kg were obtained for grains for each kilogram of available nitrogen, meaning an APUE_N_ of 122%. The effect of monoculture on yield may be associated with a greater exploration of soil volumes when the roots grow very deeply, the expression of high-affinity N transporters and a series of orchestrated changes in the metabolism for the recycling of N. (v) These results, found in the Bolivian Altiplano, lead to the search for possible strategies, such as the distribution of N in plants and possible sources of N contribution. In addition, the contribution of nitrogen from rain and the presence of microorganisms such as AMF or PGPR and endophytes favor the uptake and fixation of N in the growth of quinoa roots.

Theoretical data from this review support the hypothesis that quinoa can produce grains in soil with a lower N content, as occurs in the Bolivian highlands. For optimal performance, we also recommend that organic amendments, endophytes (bacteria and fungi), and rotation with lupines should be considered to improve N in poor soils such as those of the southern Altiplano of Bolivia.

## Figures and Tables

**Figure 1 plants-10-02479-f001:**
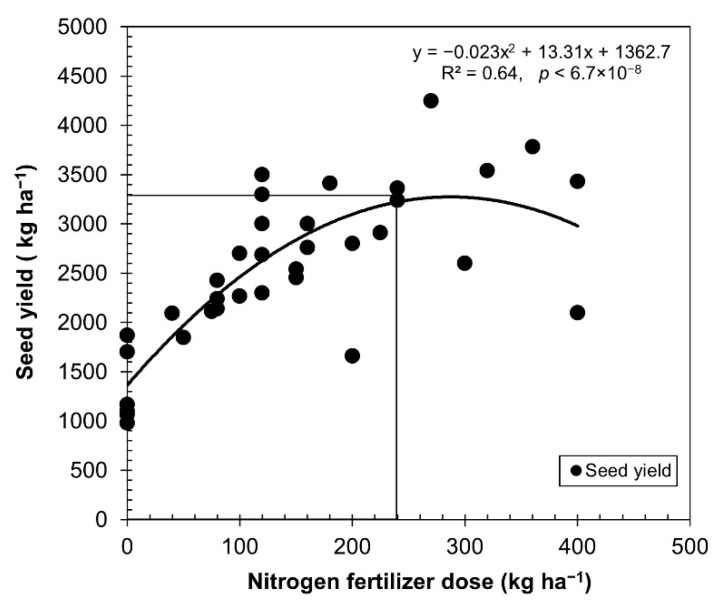
Relationship between nitrogen rates applied and seed yield. The data from Table 1 were utilized to calculate variations in seed yield in quinoa cultivated under rainfed and irrigated conditions. The symbols represent the average values of equal doses.

**Figure 2 plants-10-02479-f002:**
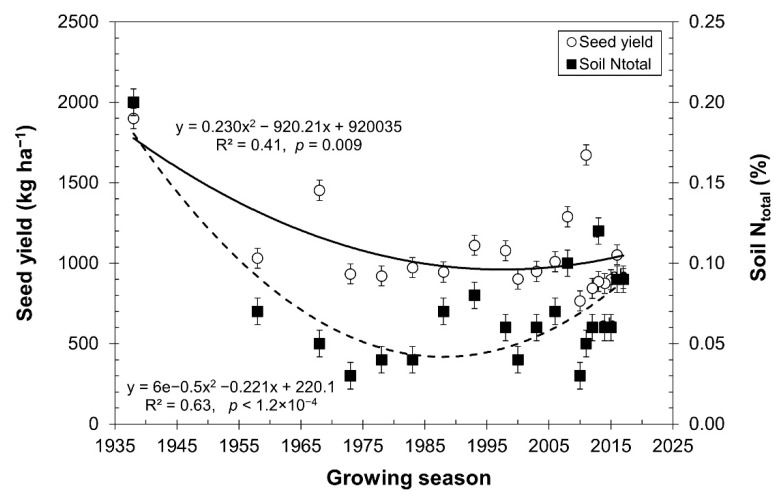
Seed yield and total soil nitrogen content from plots where quinoa monoculture has been practiced for several growing seasons (*n* = 80 years). Data adapted from Cardenas et al. [26].

**Figure 3 plants-10-02479-f003:**
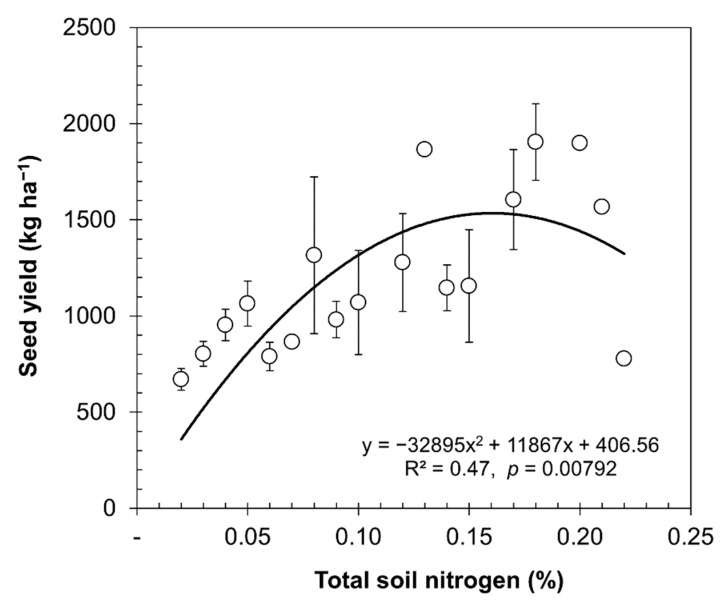
Relationship between seed yield and total soil N. The data were obtained by averaging the values of equal doses according to Table 1 (mean ± SE).

**Figure 4 plants-10-02479-f004:**
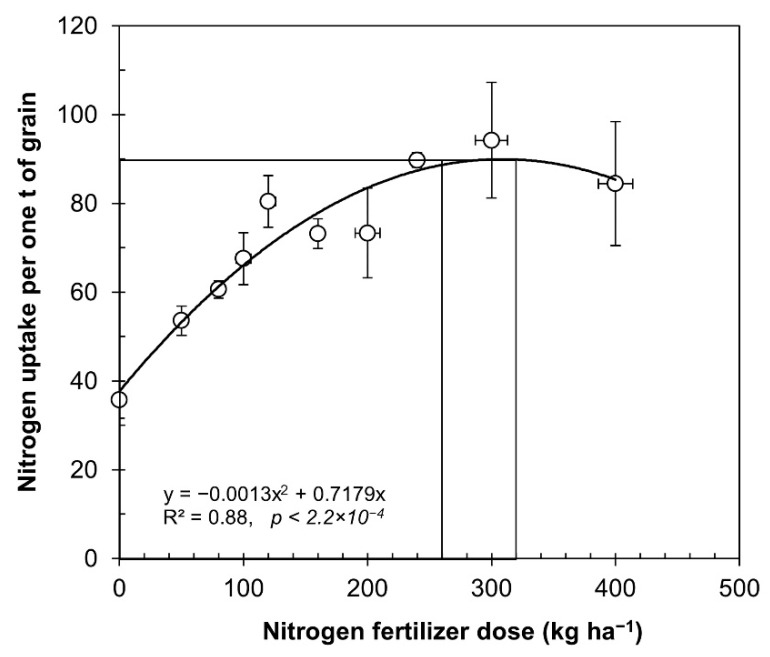
Relationship between nitrogen rates applied and uptake of nitrogen from experiments under irrigated and rainfed conditions. The data from Table 1 were utilized to calculate variations in seed yield in cultivated quinoa under rainfed and irrigated conditions. The symbols represent the average values of equal doses (mean ± SE).

**Figure 5 plants-10-02479-f005:**
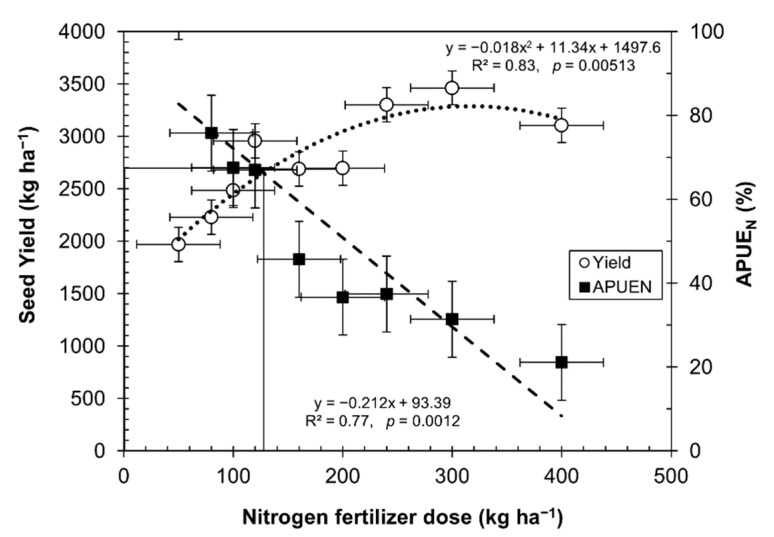
Break-even point between nitrogen fertilizer dose, seed yield, and APUE_N_ (mean ± SE). These data were obtained from Table 2.

**Figure 6 plants-10-02479-f006:**
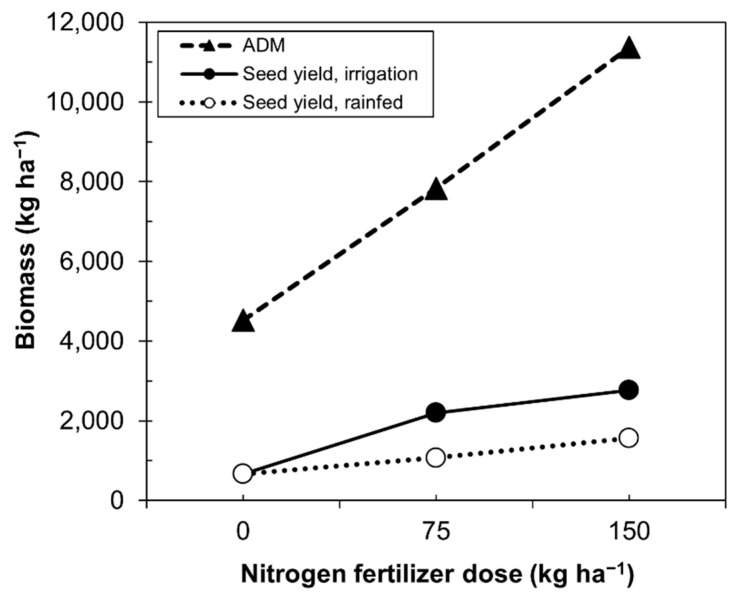
Relationship between the fertilizer dose and quinoa seed yield under rainfed and irrigation conditions. ADM, aboveground dry matter.

**Figure 7 plants-10-02479-f007:**
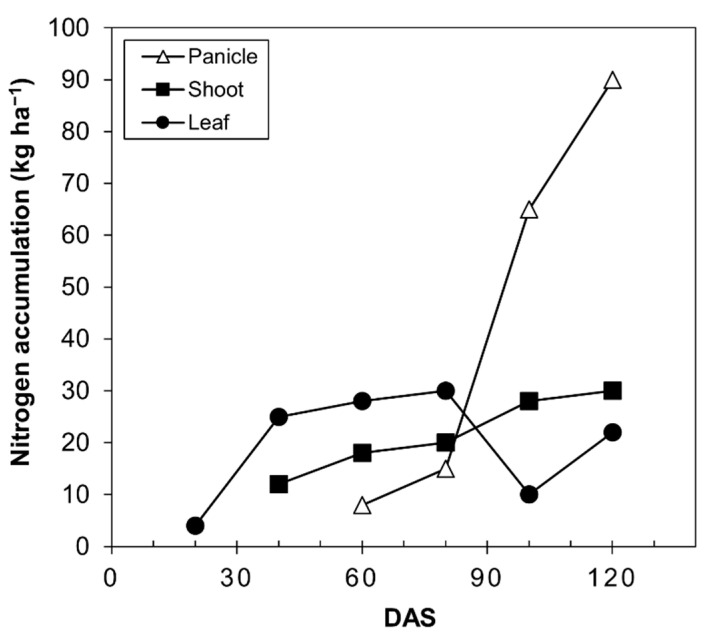
Nitrogen accumulation curve in quinoa plants (*Chenopodium quinoa* Willd) of the Imbaya variety grown under irrigated conditions. Source: Calvache and Valle [48]. DAS, days after sowing.

**Table 1 plants-10-02479-t001:** Nitrogen dose and yield of grain in various quinoa cultivars under different growing conditions and soil textures.

Dose (kg ha^−1^)	Seed Yield (kg ha^−1^)	Cultivar	Soil Texture	Irrigation Type	Reference
0	1166	Blanca Junin	Sandy clay loam–sandy loam	Rainfed	Borda, 2018 [34]
0	1100	Regalona Baer	Silty clay	Rainfed	Campillo and Contreras, 2019 [35]
40	2093	KVL 8401	Clay loam	Rainfed	Jacobsen et al., 1994 [36]
80	2428	KVL 8401	Clay loam	Rainfed	Jacobsen et al., 1994 [36]
80	2140	Regalona Baer	Silty clay	Rainfed	Campillo and Contreras, 2019 [35]
120	3500	Cochabamba y Faro	Clay loam	Rainfed	Schulte et al., 2005 [37]
120	2685	KVL 8401	Clay loam	Rainfed	Jacobsen et al., 1994 [36]
160	2760	KVL 8401	Clay loam	Rainfed	Jacobsen et al., 1994 [36]
160	3000	Regalona Baer	Silty clay	Rainfed	Campillo and Contreras, 2019 [35]
240	3360	Regalona Baer	Silty clay	Rainfed	Campillo and Contreras, 2019 [35]
320	3540	Regalona Baer	Silty clay	Rainfed	Campillo and Contreras, 2019 [35]
400	3430	Regalona Baer	Silty clay	Rainfed	Campillo and Contreras, 2019 [35]
0	1068	Faro and UdeC10	Loam–silty loam	Supplementary	Berti el al., 2000 [38]
0	1700	Altiplano INIA, Salcedo INIA	Sandy loam	Supplementary	Mendoza Nieto et al., 2016 [39]
0	1868	Blanca Real	Sandy loam	Dripping	Llaca, 2014 [40]
0	981	Genotipo O3	Loamy	Surface	Franco, 2018 [41]
50	1848	Genotipo O3	Loamy	Surface	Franco, 2018 [41]
75	2112	Faro and UdeC10	Loam–silty loam	Supplementary	Berti et al., 2000 [38]
80	2240	Blanca Real	Sandy loam	Dripping	Llaca, 2014 [40]
100	2700	Altiplano INIA, Salcedo INIA	Sandy loam	Supplementary	Mendoza Nieto et al., 2016 [39]
100	2267	Genotipo O3	Loamy	Surface	Franco, 2018 [41]
120	3300	Titicaca	Sandy loam	Deficit irrigation	Razzaghi et al., 2012 [20]
120	3000	Titicaca	Sandy clay loam	Deficit irrigation	Razzaghi et al., 2012 [20]
120	2300	Titicaca	Sandy	Deficit irrigation	Razzaghi et al., 2012 [20]
150	2456	Faro and UdeC10	Loam–silty loam	Supplementary	Berti et al., 2000 [38]
150	2541	Genotipo O3	Loamy	Surface	Franco, 2018 [41]
180	3413	Salcedo INIA	Sandy loam	Dripping	Herreros, 2018 [42]
200	2800	Altiplano INIA, Salcedo INIA	Sandy loam	Supplementary	Mendoza Nieto et al., 2016 [39]
200	1659	Genotipo O3	Loamy	Surface	Franco, 2018 [41]
225	2912	Faro and UdeC10	Loam–silty loam	Supplementary	Berti et al., 2000 [38]
240	3240	Blanca Real	Sandy loam	Dripping	Llaca, 2014 [40]
270	4249	SalcedoINIA	Sandy loam	Dripping	Herreros, 2018 [42]
300	2600	Altiplano INIA, Salcedo INIA	Sandy loam	Supplementary	Mendoza Nieto et al., 2016 [39]
360	3783	Salcedo INIA	Sandy loam	Dripping	Herreros, 2018 [42]
400	2100	Altiplano INIA, Salcedo INIA	Sandy loam	Supplementary	Mendoza Nieto et al., 2016 [39]

Source: Prepared with data from several authors.

**Table 2 plants-10-02479-t002:** Determination of the efficiencies and indices according to the total soil nitrogen content in rainfed cultivation and non-fertilized soils.

Total Soil N (%)	Available Nitrogen (AN) (kg ha^−1^)	Yield Average (kg Grains ha^−1^)	Nitrogen Harvested by Yield (kg ha^−1^)	PFP_N_ (kg Grains kg^−1^) (AN)	APUE_N_ (%)
0.02	14.9	670.3	18.2	45.1	122.8
0.03	22.3	802.9	21.8	36.0	98.0
0.04	29.7	953.9	25.9	32.1	87.4
0.05	37.1	1063.6	28.9	28.6	77.9
0.06	44.6	789.2	21.5	17.7	48.2
0.07	51.9	865.5	23.5	16.7	45.3
0.08	59.4	1316.0	35.8	22.2	60.3
0.09	66.8	948.6	25.8	14.2	38.6
0.1	74.2	1070.4	29.1	14.4	39.2
0.12	89.1	1277.9	34.8	14.3	39.0
0.13	96.5	1865.9	50.8	19.3	52.6
0.14	103.9	1145.6	31.2	11.0	30.0
0.15	111.4	1156.1	31.4	10.4	28.2
0.17	126.2	604.9	43.7	12.7	34.6
0.18	133.7	1904.0	51.8	14.2	38.7
0.2	148.5	1899.2	51.7	12.8	34.8
0.21	155.9	1567.3	42.6	10.1	27.3
0.22	163.4	777.3	21.1	4.8	12.9

Source: Elaborated from Cardenas et al. [26].

**Table 3 plants-10-02479-t003:** Efficiency indicators according to various fertilization tests under irrigated and rainfed conditions.

Nitrogen Fertilizer Dose (kg ha^−1^)	Average Yield (kg Grains ha^−1^)	Nitrogen Harvested by Yield (kg ha^−1^)	Partial Factor Productivity of Nitrogen (PFP_N_) (kg Grains kg ^−1^) (AN)	Apparent Use Efficiency of N (APUE_N_) (%)
50	1848	50.3	37.0	100.5
60	1771	48.2	29.5	80.3
75	2112	57.4	28.2	76.6
80	2314	62.9	28.9	78.7
100	2483	67.5	24.8	67.5
120	2749	74.8	22.9	62.3
150	2453	66.7	16.4	44.5
160	2882	78.4	18.0	49.0
180	3413	92.8	19.0	51.6
200	2193	59.6	11.0	29.8
225	2912	79.2	12.9	35.2
240	3300	89.8	13.8	37.4
300	2600	70.7	8.7	23.6
320	3540	96.3	11.1	30.1
360	3783	102.9	10.5	28.6
400	2765	75.2	6.9	18.8
Average	2695	73.3	19.7	50.9
*R* ^2^	0.83 *	0.88 **	0.83 **	0.77 *

Significant at the * 0.05 and ** 0.01 probability levels. Source: Calculated from Table 1.

**Table 4 plants-10-02479-t004:** Effect of nitrogen fertilizer dose application on the production of aboveground dry matter (kg ha^−1^) in three quinoa varieties grown under irrigated conditions in Ecuador.

DAS	20	40	60	80	100	120
Nitrogen Fertilizer Dose (kg ha^−1^)						
0	166.6	712.7	1407.3	1835.0	3967.5	4524.8
75	183.4	948.9	2226.8	3650.8	7065.4	7832.9
150	221.7	1055.7	2659.7	5002.4	9943.7	11,366.1

Source: Adapted from Calvache and Valle [48]. DAS, days after sowing.

## Data Availability

The data presented in this study are available in this article.

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
