# Peer review of "Quinoa (Chenopodium quinoa Wild.) Seed Yield and Efficiency in Soils Deficient of Nitrogen in the Bolivian Altiplano: An Analytical Review"

_plants, 2021, doi:10.3390/plants10112479_

Round 1

Reviewer 1 Report

Authors tried to improve the manuscript but unfortunatelly I do not think they were succeeded. Most implementations are rather difficult to understand. Language requires a major improvement by an English native. Authors took some sentences from literature and used them, sometimes less precisely. But again, for me, the main issue with this manuscript is the less correct definition(s) of nitrogen use efficiency. I am sorry because I misunderstood the study country. 

Authors proposed a Review article but it is not well demonstrated if the article is about the nitrogen use efficiency (there are several manuscripts, including review articles, on this topic) or the quinoa. Authors did not develop strongly the quinoa interest and its N use efficiency. I am not sure if authors consider this crop a legume, but it is a pseudocereal. In the Introduction section, authors dedicated several paragraphs to N cycling, which is not relevant for the present study, and some parts are not correctly presented. 

Because quinoa can be considered a halophyte plant (resistent to the salinity), where the old and young leaves accumulate Na+ but are both functionally efficient under saline conditions, this could be an interesting aspect explored in the present review.

Some specific comments:

Keywords should be improved: write: "low fertility soil; quinoa efficient crop; arid environment..."

Introduction:

-L31: delete "ensure"

-L36: replace "inorganic N" by "reactive N", and improve the sentence.

-L38: N forms in water were not mentioned in L34/35

-L44: do you mean "soil system"?

-L46/47: you could delete the sentence

-L36/52: I think you could delete. N biochemical cycle is well-known and is not directly linked to the present study. The comments on nitrogen use efficiency (NUE) by plants is of poor quality. 

-L53: NUE in soils? Or by plants?

-L58: add "availability of N in soil"

-L59: "transition between assimilation and recycling of N". Please, clarify

-L61: add "...in the plant."

-L63: delete "plants". You should not use both terms "crop plant" altogether.

-L63/67: this concept is more an "Agronomic N use efficiency". NUE is normally related to the amount of fertilizer N taken by the crop. I disagree from Reference 17. Delete.

-L82: delete "water stress"

-L104: "and how N cycle fulfilled is?". What do you mean?

-L107: replace "Compared to" by "On the contrary,"

-L111/112: move to L106

-L114: the value indicated for protein concentration (23%) seems too high, according to the literature (about 16%).

-L121/123: this is not appropriately addessed. NUE is not directly a function of soil microorganisms and atmospheric N2 in non-legumes, only if you consider the free living N2 fixing bacteria. The crop is a cereal-like plant.

-L125: "without applications". What do you mean?

Results:

-L134/136: this is not a correct explanation. What about the plant characteristics?

-Fig. 1: as another Reviewer mentioned, "Extraction rate" is not correct. This seems more a NUE.

-L139: improve the sentence

-L147/150: this is not true, according to Fig. 1. The maximum yield was not obtained for 120 kg M ha-1. The same comment for L154/155.

-Table 2: the 3rd column should refer to NUE, and apparently represents the same as the 4th column, therefore data should be the same which is not the case.

I could not go further with the revision of this manuscript because the issues are so many.

Certain References used are not the most appropriate.

Author Response

All the commnents of reviewer have been accepted. The manuscript has undergone languaje revision (see certificate). I appreciatte the valuable comments that have improved and enriched this paper.

Reviewer 2 Report

The authors accepted suggested corrections and significantly modified and improved the manuscript. Abstract needs some improvement, and a little smoothing; the part reads everyone. 

There are several more errors or mistakes to be corrected and improve the quality of the manuscript.

Still, I do not understand why Materials and Methods chapter are placed after Results and Discussion ones. A reader needs to read methods, first, to understand your approach to analysis.

Comments

l.15 Just exactly this year? (also do not agree with many research articles (Tab. 1), the corresponding citation in Introduction points to some publication FAO, without any additional information. 

l.19 ...Andean grain...? Rather, grain crop

l.19-20 A bit clumsy sentence, could be formulated better

l.21 Of course, 120 kg is not an exact limit, should be a bit  explained,....the analysis suggests ...on average...  

l.23 Under N limiting conditions the uptake is hardly massive – use, e.g., .....key factors for sustaining N uptake under limiting....

The last two sentences should show the conclusions are based on your extensive review of relevant literature, e.g. .....Our study/review/analysis of literature shows/suggests........

l.66 ...remove... rather, uptake, extract, deplete,...

l.81 ... and due to/thanks to its adaptability...

l.121 It is still unclear for readers - farmers uproot and (probably) and take away/remove plants from a field

l.138 Why only mutant cultivars?

l.148, Fig. 1 Add regression equation to graph, to confirm the inflection point

Tab.2 Again, the fertilization dose is mentioned in the text (l. 151) referring to Tab. 2, however, here it is denoted as “Available nitrogen” – it is very confusing (also l. 176); in Tab.3 Nan denotes different thing. If both, data on fertilization rates and soil available N (calculated or determined) are mixed, it should be mentioned/stressed in the text. 

l.155 After this point (120 kg N) yields DO NOT decrease, it rather 300 kg N (Fig. 2), also l. 530

l.191-192 Clumsy statement

l.197 Unclear sentence; further, using present time you generalise the results of one experiment.

Fig. 3 I do not understand the sense of showing significant differences, here, what do you try to prove? Add, “Soil Ntotal”  to caption or axis description (like in Fig. 4)

l.272 Only Extracted N is shown in Fig. 1, not seed yield. In Fig. 2 the yield decreases at about 300 kg N dose

l.291 Again, not only vegetables but plants

l.311-313 Correct While...while 

l.338 nitrogen x Nitrogen

l.338-9 .....nitrogen content.....associated with similar studies??

l.367 Using ...nitrate...  without context in the part of manuscript is confusing, use at least ....soil nitrate content

l.505 In Tab. 1 data of PFPN derived either from fertilization rates, or from soil available N should be marked

l.511 Add “soil” total nitrogen.... here and elsewhere, otherwise, it is confusing for readers

l.515  “cc” unit is unclear, use “g cm-3”

I m not native English speaking but it seems in some sentences the subject is missing (e.g. l. 16 ...also ? is considered ..., l. 25....plant NUE, and ? is a new approach, l. 86, l. 107), further, there are small grammar mistakes throughout the manuscript, please, check the text once again (example: l. 423 “......argine increase” Should be increases or increased.

Author Response

(The authors gave the same response as above.)

Reviewer 3 Report

Title: Quinoa Seed Yield and Nitrogen Efficiency in Poor Soils of the Southern Bolivian Altiplano: An Analytical Review

# The affiliations of the authors should be internationally adopted language # What does poor soil mean?????? Please clarify

 # Abstracts should be clear, according to the title, with scientific information regarding whatever you are covering in this article.

# With the English edition, you need to improve the clarity of sentences and compositions of the introduction.  

# Need to work on abbreviations and avoid repetition

# Methodology should be clearer with proper references

#it is best to simplify the equation and avoid repetitious words

#Discuss is poorly written, you should begin with your own figures / table and then with the rest of the workers. 

# Conclusion based the article and in built points with a par

Overall, the article needed major revisions with fine-tuned English 

Author Response

(The authors gave the same response as above.)

Reviewer 4 Report

An important work concerning an unusual plant species. Please take care of english language in the whole text. A native english speaker should take care of it. My comments are included in the attached pdf file.

Author Response

(The authors gave the same response as above.)

Reviewer 5 Report

Article “Quinoa Seed Yield and Nitrogen Efficiency in Poor Soils of the Southern Bolivian Altiplano: An Analytical Review”, ID 1373924

Abstract should be a mini-version of the article and contain information from all the sections of the article. This Abstract has only Introduction and Results and Conclusion part. Nothing from the Materials and Methods section is included.

Introduction. I strongly recommend to mention Latin scientific names of all living organisms (plants, animals, microorganisms), when mentioning them for the 1st time, staring with quinoa in line 55 and following with those in lines 90, and 106–108. Term ‘maize’ is recommended instead of ‘corn’.

The sentence should not start with a numeric number (line 101 in Introduction; see also other places in the article, e.g. line 339).

Materials and Methods.

The description in this section is insufficient. For example, in line 501, it is mentioned that (quotation) “Finally, 20 articles were considered to use the data for fertilization levels”.  I would like to see the numbers of the 20 sources (articles) used. In addition, while reading the article I see that data from used sources is mathematically processed: mean±SE, R2, p-values are shown in Figures, average values are shown in Table 2 (quotation: “Table 2 shows efficiency indicators based on the average yields per fertilization dose, according to data from average values in Table 1” (lines 150–151)); unfortunately, description of used mathematical methods and programme-package is not available in this section.

SI system’s symbols of units should be used: I guess that used symbol ”cc” (line 515) means  cm3, if “yes”, please, use it, if “no”, please explain what does this “cc” means?

Results.

Description in this section raises several questions and is incomprehensive in several places.

The title of Table 1 is not complete; Table shows much more than the title promises. I suggest to specify the title. Maybe I do not understand the family-name writing in Spanish, but references in Table 1 show only part of family name that appears in the Reference List, e.g., Nieto et al., 2016 [48] (in Table 1) is written as Mendoza Nieto in reference List.

Lines 139–140: it seems that unnecessary full stop appears in the middle of the sentence (after the words Table 1)???

In several places it is not possible to understand when the authors describe data obtained by their calculations and when – simply data from literature sources. E.g., they tell what is shown by Figure 1 (lines 140–141), then Figure itself appears, and I suppose that further description of Figure follows, but description is about yield which increases up to the nitrogen dose 120 kg ha-1 (lines 147–150); I do not see it in Figure 1 in which other parameters are shown. In lines 165–169 it is written (quotation) “According to Table 2 and Figure 2, the extraction increases as the yield increases. However, this does not correlate with higher efficiency, the data show a higher efficiency in the use of nitrogen in plants grown in soils with very low nitrogen levels”. I do not see any extraction shown in Figure 2; seed yield is shown on the y-axis, and nitrogen dose on the x-axis. Directly after that it is written (lines 168–169): “Three doses of nitrogen fertilization (25, 50, 100 kg N ha-1) were used and the extraction was 25 kg N per ton of grain produced (1:40 ratio) [61]”. In Table 2 and Figure 2 data start with the dose of 50 kg N ha-1, and not with the 25 kg. Reference is put, of course [61], but ideas from tables and figures merge with the ideas from literature sources and it is too confusing. Confusing is also the text in lines 171–198 and 245–246 where it is not always understandable from where the data came. Description must be clearer.

In lines 203–205 it is promised “Figure 3 shows the intensity of land use (years) and the N available for the 2016–2017 season at the intersalar area in the southern Bolivian Altiplano [32].” When I look at Figure 3, I see years from 1930 to 2030 on x-axis. Description needs to be more precise.

Authors have to describe symbols and lines used in all Figures more precisely. 

The numbers in which the dot appears twice are incomprehensible (Table 3), e.g., 1.063.6; as I know, thousands should be separated by a comma (1,063.6).

What does the reference “Elaborated from Unpublished data Cardenas et al. [32]” (e.g., line 232) mean? When I look at the list (lines 624–625), it looks like the work has been published as described.

I do not see yield stabilization in Figure 5 (line 271; Figure 5 in page 10)

In my opinion, sentence in lines 304–305 is not complete (quotation): “After the plant takes the nitrogen and metabolizes it into plant proteins.”

The capital initial letter is used in some places in the text for words which are not proper nouns, e.g., Nitrogen (line 338), Glutamine (line 364), Betalains (line 365) etc.

Incomprehensible sentence appears in lines 350–351 (quotation): “Other research conducted on quinoa by [26], shows that the absorption of N almost doubled with fertilization of 120 kg N ha-1 compared to N”.

Conclusions. Conclusions are well-written and give answers to the questions set up at the end of Introduction, but I would like to see these answers in clearer, more comprehensive description in Results section.

I suggest writing full words PFPN, AMF and PGPR in the section “Conclusions”, as the available nitrogen (AN) is already written (lines 528, 535).

References. High amount of references was used. I did not find a demand in instructions for authors that titles of articles written in other language than English have to be translated into English and put in square brackets after the title in original language. But for me, it was inconvenient that I could not even understand the topic of the article referred by authors in 37 cases. In my opinion, an international-reader-oriented, comprehensible list should be established.

Reference list should be checked once more because, e.g., description of the source [81] (line 734) is not complete.

Author Response

(The authors gave the same response as above.)

Round 2

Reviewer 1 Report

My decision is Major revision according to:

Sub-section 3.3.: authors should replace "assimiliation" with "assimilation"

Figures 6 and 7: authors should verify the legends

Section 3.5: authors must delete. This is not relevant and is only based on hypotheses.

Conclusions: auhors must improve. Conclusions are not a summary.
They should indicate the main results. Therefore I would recommend that authors should delete from "On the contrary...." till the end.

Author Response

AUTHORS: the manuscript was sent for review of the English language (see certificate)

In this review, the authors have tried to accept all the observations of the five reviewers, whom we thank for their valuable contributions that have contributed to substantially improve the manuscript. All responses are in blue. Thank you very much to all.

Comments and Suggestions for Authors

My decision is Major revision according to:

Sub-section 3.3.: authors should replace "assimiliation" with "assimilation"

AUTHORS: corrected. Thanks for your patience. (4th November 2021)

Figures 6 and 7: authors should verify the legends

AUTHORS: was verified and corrected and improved the figure captions. (4th November 2021)

Section 3.5: authors must delete. This is not relevant and is only based on hypotheses.

AUTHORS: this section is not based on hypotheses. We propose these solutions based in field work and research knowledge. (4th November 2021)

Conclusions: auhors must improve. Conclusions are not a summary.
They should indicate the main results. Therefore I would recommend that authors should delete from "On the contrary...." till the end.

AUTHORS the conclusions are improved according to your suggestions and the others reviewers. Many thanks. (4th November 2021)

Globally, quinoa cultivation is of interest due to its versatility, resilience, and provider of a nutritious grain. Nevertheless, quinoa cultivated in Andean countries, particularly in Bolivia are facing several constraints as low fertility in soils. Data obtained from this review provided a better knowledge about increasing yield of quinoa (i) the optimal amount under these conditions was determined to be 240 kg of N with 3600 kg grain ha–1 (ii) the fertilization limits were obtained from the intersection of APUEN and seed yield, finding that the equilibrium point appeared to be up to 130 kg ha–1 of nitrogen, which is enough to produce 2700 kg ha–1, (iii) the nitrogen uptake and assimilation parameters presented an inverse relationship, with higher doses of fertilizers showing a lower efficiency of nitrogen utilization in grain production. The information obtained shows that in the event of an increase in nitrogen fertilization, an important part of the N is destined for vegetative growth. (iv) Under monoculture practice without nitrogen fertilizers, as occurs in the Bolivian Altiplano, the response found in terms of yields is similar to the practice of irrigation and fertilization, although in much smaller proportions, in soils with very low nitrogen (0.02% total N), equivalent to 14.9 kg ha–1 N available. Meanwhile, 670 kg grains ha–1 and 45.1 kg were obtained for grains for each kilogram of available nitrogen, meaning an APUEN of 122%. The effect of monoculture on yield may be associated with a greater exploration of soil volumes when the roots grow very deeply, the expression of high-affinity N transporters and a series of orchestrated changes in the metabolism for the recycling of N. (v) These results, found in the Bolivian Altiplano, lead to the search for possible strategies, such as the distribution of N in plants and possible sources of N contribution. In addition, the contribution of nitrogen from rain and the presence of microorganisms such as AMF or PGPR and endophytes favor the uptake and fixation of N in the growth of quinoa roots.

Theoretical data from this review support the hypothesis that quinoa can produce grains in soil with a lower N content, as occurs in the Bolivian highlands. For optimal performance, we also recommend that organic amendments, endophytes (bacteria and fungi), and rotation with lupines should be considered to improve N in poor soils such as those of the southern Altiplano of Bolivia.

Reviewer 3 Report

Revision: Minor 

Title: Quinoa (Chenopodium quinoa Wild.) Seed Yield and Efficiency in Soils Deficient of Nitrogen in the Bolivian Altiplano:  An Analytical Review

Abstract: Nothing is a novelty in the abstract, it is presenting collected generic information, it should be clear about the message to the scientific society.  

Introduction   

#Need to do a lot of work on abbreviations

# No background information is present about the topic in the introduction with data, as per my earlier comments.  

# Hypothesis of the topic will be clear in the abstract, introduction and conclusion present the massage clearly about the article.

# Table description with the justification of the data may be present.

# As per my earlier comments conclusion should be inbuilt points in built points and provide a clear message to the international reader.

Overall, the information is good but need a lot of work sentence connectivity, scientific justification on the tables with English improvement.

Author Response

AUTHORS: the abstract was improved and thanks for your valuable recommendation. (4th November 2021)

Current abstract:

ABSTRACT: Quinoa is a strategic crop due to its high N content and its adaptability to adverse conditions, where most of the soils are deficient of nitrogen (N). The central question in this review was: How can quinoa yield low levels of nitrogen in the soils of Altiplano? This question was unraveled based on different factors: (1) fertilization effect on productivity, (2) fertilization limits, (3) uptake and assimilation of nitrogen parameters, (4) monoculture practice effect, and (5) possible sources and strategies. One hundred eleven articles of different scientific platforms were revised and data were collected. Information from articles was used to calculate the partial factor productivity for nitrogen (PFPN), the apparent use efficiency of N (APUEN), available nitrogen (AN), and nitrogen content harvested in grains (HarvN). Quinoa responds positively to fertilization, but differences in yield were found among irrigated and rainfed conditions. Quinoa can produce 1850 kg grains ha−1 with 50 kg N ha–1 under irrigated conditions, and 670 kg grains ha–1 with 15 kg N ha–1 in rainfed conditions. Quinoa increases seed yield and HarvN while increases N fertilization, but decreases nitrogen efficiency. In Altiplano, without nitrogen fertilizer, the quinoa yield relies on between 500 and 1000 kg ha−1, which shows that in the soil, there are other nitrogen sources. (4th November 2021)

Introduction   

#Need to do a lot of work on abbreviations

AUTHORS: abbreviations were described. Thanks for your patience. (4th November 2021)

# No background information is present about the topic in the introduction with data, as per my earlier comments.  

AUTHORS: we included. (4th November 2021)

# Hypothesis of the topic will be clear in the abstract, introduction and conclusion present the massage clearly about the article.

AUTHORS: they were cleared and included in this sections. (4th November 2021)

# Table description with the justification of the data may be present.

AUTHORS: tables are well justified, and the description were improved. (4th November 2021)

# As per my earlier comments conclusion should be inbuilt points in built points and provide a clear message to the international reader.

AUTHORS: conclusions were improved and described as per your recommendation. (4th November 2021)

Overall, the information is good but need a lot of work sentence connectivity, scientific justification on the tables with English improvement.

AUTHORS: the English was edited by the professional service of MDPI. (4th November 2021)

Reviewer 4 Report

You have positively corresponded to reviewers comments.

Author Response

AUTHORS: the manuscript was sent for review of the English language (see certificate)

In this review, the authors have tried to accept all the observations of the five reviewers, whom we thank for their valuable contributions that have contributed to substantially improve the manuscript. All responses are in blue. Thank you very much to all.

AUTHORS: Many thanks for your positive assessment. (4th November 2021)

Reviewer 5 Report

The article appears to have been refined in line with the main recommendation. I still didn't understand the small nuance - has the measurement unit 'cc' corrected or explained? Ido not know such a measurement unit in SI System.

Author Response

AUTHORS: Sorry for this forgotten detail. We have corrected and added in the manuscript. The unit is cubic centimetre or cm-3. Thanks for your patience. (4th November 2021)

AUTHORS: the manuscript was sent for review of the English language (see certificate)

This manuscript is a resubmission of an earlier submission. The following is a list of the peer review reports and author responses from that submission.

Round 1

Reviewer 1 Report

Topic is relevant dealing with nitrogen use efficiency by quinoa grown in Colombia. The study uses metadata taken from literature.

The concept of nitrogen use efficiency is not well addressed in the manuscript. Therefore, several issues are observed in Results presentation and Discussion which is not convincingly provided. Besides, to estimate the nitrogen use efficiency, authors did not consider a control (N0), therefore the experimental layout has some limitations.

Figure 4 is probably misunderstood with Figure 3.

Reviewer 2 Report

The comments are in attached pdf.
